# Effects of the Transverse Deck-Roadbed Pounding on the Seismic Behaviors of the Prefabricated Frame Bridge

**Yuwei Wang [1], Jinli Zhang [1], Yingao Zhang [2], Rui Zuo [2], Liang Chen [2] and Tianyue Sun [2],***

[1]   Anhui Transportation Holding Group Co., Ltd., Hefei 230088, China
[2]   Department of Civil Engineering, Hefei University of Technology, Hefei 230009, China
*    Correspondence: 2019170551@mail.hfut.edu.cn

**Abstract:** Pounding effects on prefabricated frame bridges are not clear, which may influence seismic behaviors a lot in some cases. Prefabricated frame bridges are emerging structures designed to solve the problem of difficult land acquisition in highway expansion and reconstruction, the deck of the prefabricated frame bridge is adjacent to the original roadbed in the transverse direction, so the pounding potential exists under the earthquake ground motions. In this study, the artificial ground motions of the different seismic intensities are selected to carry out the nonlinear time history analyses, and the pounding effects on the prefabricated frame bridge are evaluated based on the pounding forces and the components' seismic response. It is found that the pounding effects are not obvious in all cases; some energy can be dissipated in the pounding process, which is also limited to some extent. Finally, the influences of the gap distance and seismic intensity are investigated according to the parameter sensitivity analysis. The results indicate that the gap distance and the seismic intensity are the two important factors related to the pounding effects, the seismic response of the components will decrease when the pounding effects are obvious, and the transverse deformation of the deck cannot influence the stress state of the superstructure.

**Keywords:** pounding; prefabricated frame bridge; seismic behaviors

## 1. Introduction

Prefabricated frame bridges are emerging structures designed to solve the problem of difficult land acquisition in highway expansion and reconstruction, which is based on the strategy of sustainable development and environmentally friendly construction [1]. As shown in Figures 1 and 2, the prefabricated frame bridge is very long, with 15 spans of 6 m each. The superstructure of the prefabricated frame bridge is different from the ordinary bridge girder, which is designed to be very thin. Some mild rebars are installed in the deck, but no pre-stressed strands are installed. Piers and piles all consist of prestressed columns [2]. In terms of the connection modes between the girder and the piers, bearings are usually applied on traditional bridges to dissipate seismic energy and restrict the deformation of the deck [3,4]. Considering that bearing maintenance is a hard task and a large number of bearings are needed, which costs a lot [5], an elastoplastic column-deck joint (EC-DJ) is developed to replace bearings to connect the deck and the prestressed columns. The structural style of the prefabricated frame bridge is similar to the wharf in a sense [2,6].

Pounding force is a complicated load under seismic excitation, which occurs in a short time. Many factors influence the pounding potential between the adjacent segments, such as friction, gap distance, ground motion intensity, structure stiffness, and segment stiffness, so the process of energy transfer is complex [7,8]. Earthquake-induced pounding tends to be detrimental to the structures, which may cause material stiffness degradation, sliding, and even girder collapse [9,10]. Furthermore, the seismic behaviors of the adjacent structures can be affected to varying degrees by the pounding effect under different conditions [11,12].

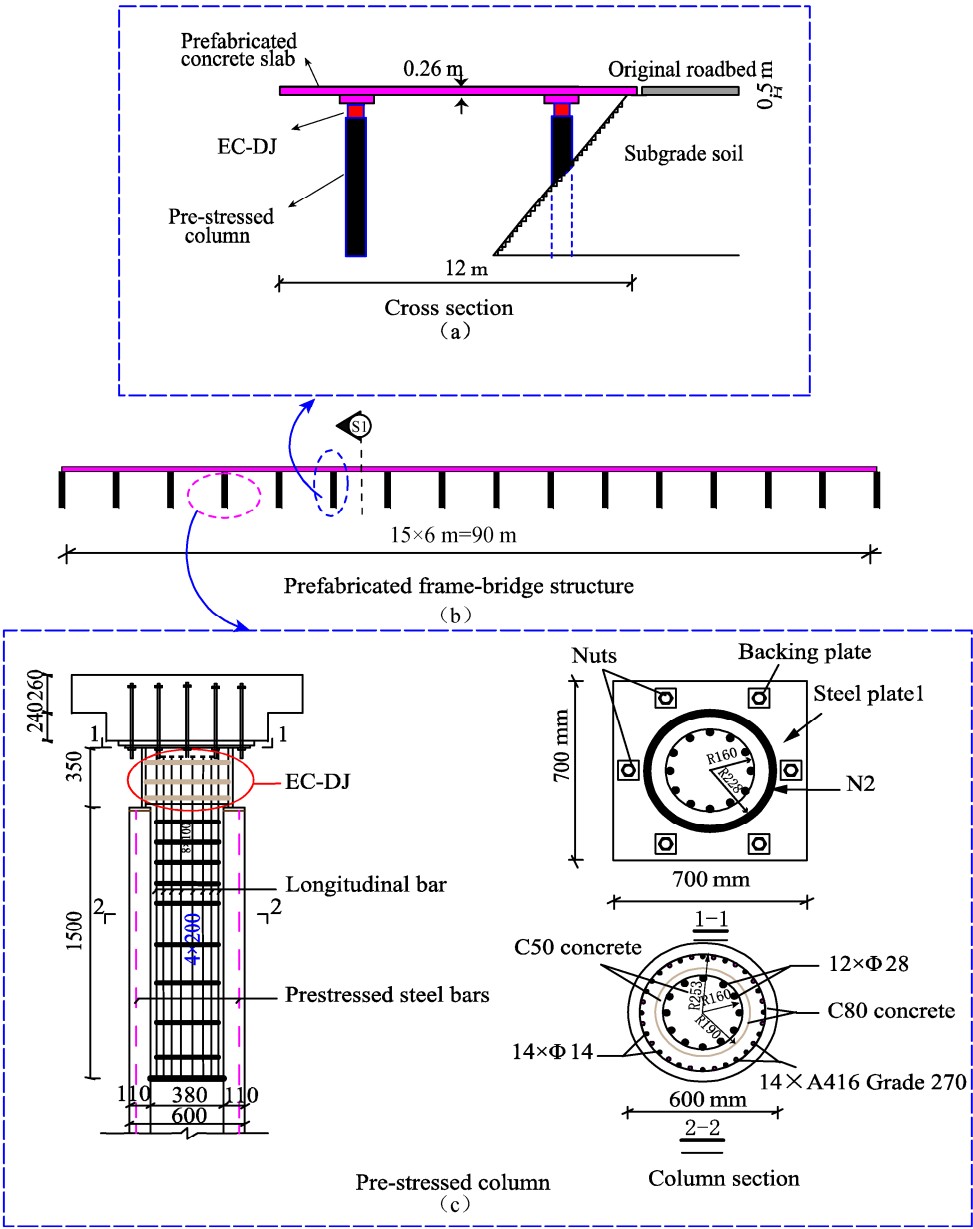

**Figure 1.** The configuration of the prefabricated frame bridge. (**a**) Cross section. (**b**) Pre-fabricated frame bridge structure. (**c**) Pre-stressed column.

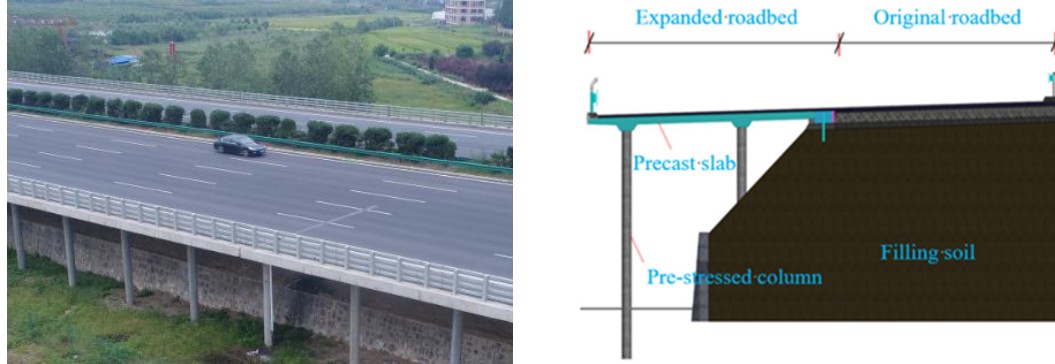

**Figure 2.** The prefabricated frame bridge structure.

Many studies have been conducted to investigate the pounding effects on bridges under earthquake action. The pounding in the longitudinal direction was mostly between the girders and abutments or between adjacent girders, while poundings in the transverse direction were mostly between the retainers and the girders [8,13,14]. Some scholars also studied the bidirectional poundings of skew bridges and curved bridges [15,16]. Li B. et al. [17] found that the pier's response decreases with the consideration of the pounding force, and the pounding potential increases with the stiffness of the larger movable abutments. Chouw N. et al. [18] maintained that when the soil-structure interaction and ground motion spatial variation are considered, the pounding damage potentials between the adjacent decks may not be reduced under high seismic intensity. Some scholars proposed methods for mitigating the pounding damages. Guo A. et al. [13] present a magnetorheological damper on reducing the pounding of the adjacent superstructures. Abdel Raheem S.E. et al. [19] used a shock absorber to mitigate the pounding, so the pier forces could be smoothed due to the reduction of the pounding forces. The seismic performance can also be improved by retainers [14,20], which can restrict the transverse deformation of the girder due to pounding.

In seismic design, pounding potential is usually an important factor that affects the analysis results. Kun C. et al. [16,21] suggested that the seismic responses are significantly underestimated when the pounding effect is ignored for the skew bridges. Li B. and Chouw N. [22] investigated the impacts of spatial variation of ground motions on the pounding based on the shaking table and concluded that pounding damage may be overestimated for the elastic bridge studies. Huo Y. et al. [23] maintained a similar view for the skew bridges and provided valuable guidance for future bridge design.

However, in some cases, poundings can consume some seismic energy and limit the displacement of the structure, which is conducive to improving the seismic performance of the structure [13,24]. The prefabricated frame bridge is a new structure that is constructed based on the original roadbed to expand the road. The deck is likely to pound with the roadbed due to the transverse deformation under the seismic action, but the advantages and disadvantages of this to the overall seismic performance of the structure are unknown.

In this study, the artificial ground motions are generated by the SIMOKE code first, which are selected as the seismic excitation in the time history analysis. Then the seismic response law of the prefabricated frame bridge affected by the pounding effect is investigated. Finally, the influences of the interaction of the gap distance, the transverse deformation, and the ground motions intensity are considered to explore the pounding effect on the prefabricated frame bridge. The results indicate that the pounding between the deck and the roadbed can be ignored in cases where the intensity of the ground motions is between 0.2 g and 0.5 g. When the seismic intensity is moderate, seismic energy can be consumed, and the transverse deformation can be restricted obviously due to the pounding effect. Meanwhile, the response of the structure will also be enlarged with the decreasing gap distance and the increasing intensity of the pounding. According to the nonlinear time-history analysis and the parameter sensitivity analysis, the possible impact of the pounding on the prefabricated frame bridge is clear. Based on the analysis, some proper suggestions and evaluations for the seismic design are provided, which provides a reference for future research.

## 2. Prefabricated Frame Bridge Structure and Finite Element Model

### 2.1. Layout of the Prefabricated Frame Bridge

Prefabricated frame bridges are emerging structures for the reconstruction and expansion of the expressway. As shown in Figures 1 and 2, the fifteen-span prefabricated frame bridge has a total length of 90 m and a width of 12 m, with spans of 6 m. The superstructure of the prefabricated frame bridge is comprised of decks, and the substructure is comprised of pre-stressed columns. An elastoplastic pile-deck joint (EP-DJ) is applied to connect the decks and the pre-stressed columns, which is comprised of the steel tube, confined concrete, and stirrups, as shown in Figure 1c. The thickness of the deck is 0.26 m, and the hunch is

0.24 m. The pre-stressed column is 9 m in height; the diameter is 0.6 m in the outer and 0.38 m in the inner. The depth of the column body buried in the soil is set as 3 m.

### 2.2. Structure Analysis Model

The dynamic finite element model was built using the OpenSees analysis software [25], in which the material nonlinearity is considered, as shown in Figure 3. Rigid connections were used between the prefabricated decks and EC-DJs. The nonlinear behavior of the pre-stressed columns was simulated by the nonlinear beam-column element, and the interactions between columns and soil were considered, which was simulated by a zero-length element. The shell elements were used to simulate prefabricated decks, which are expected to remain elastic under the earthquake actions, and the pounding effects between the decks and roadbed are considered. The pounding model uses the linear spring model, and the energy dissipation is considered by the damping material. The roadbed and the foundations are regarded as fixed boundaries. The material models are shown in Figure 3c,d, and the detailed introduction of the materials is referred to the reference [2,26,27].

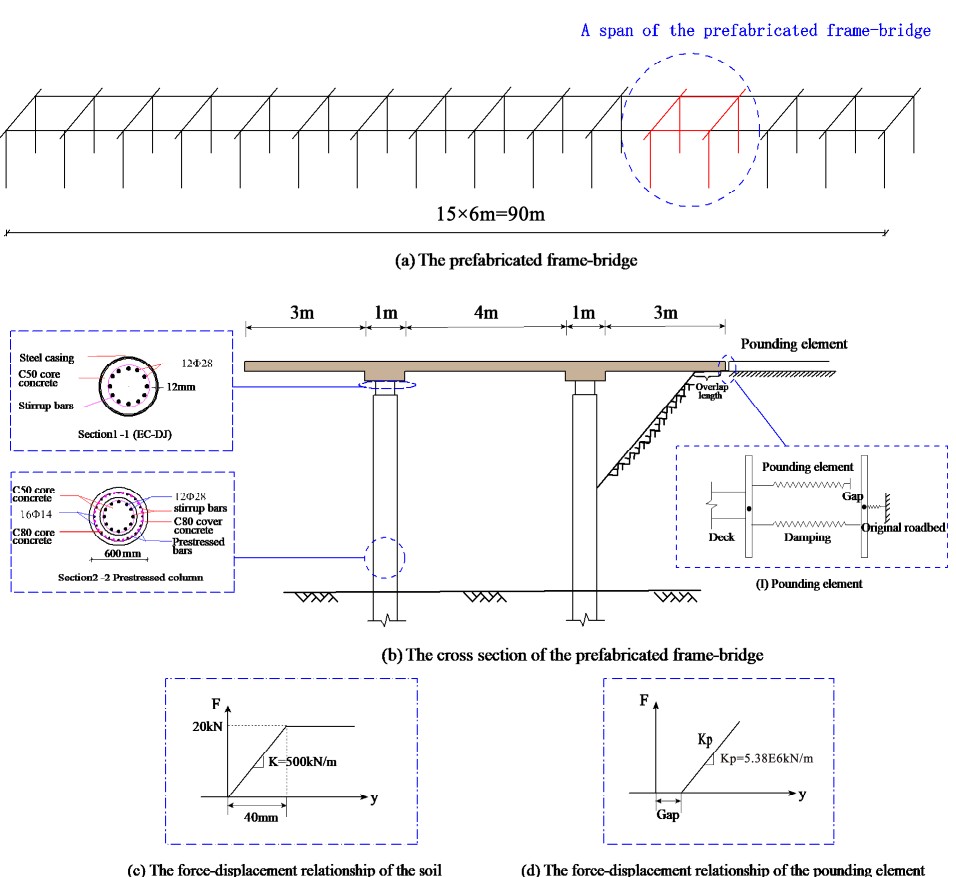

**Figure 3.** A span of the prefabricated frame bridge.

### 2.3. Consideration of the Pounding Element and the Soil

When the prefabricated frame bridge structure is under seismic excitation, passive earth pressure will be generated from roadbed soil, and the transverse deformation of the deck will be restricted. In this process, pounding force is generated, which influences the response of the structure to some extent. In this study, a bilinear model is applied to simulate the constitution of the roadbed soil, as shown in Figure 3. The initial stiffness is set as 500 kN/m, and the yield force is set as 20 kN [28,29].

Jankowski [30] maintained that the Kevin model can reflect the pounding effect accurately if the proper parameters are selected. In this study, the pounding effect between the deck and the roadbed is simulated by the Kevin model [31–33], which is composed of

linear springs and dampers in parallel. The pounding force is simulated by the linear spring element, and the consumed energy is also considered in the model, which is simulated by the damping material.

Moreover, the pounding force can be calculated as follows:

$$F(y) = \begin{cases} K_P(d_0 + y) & d_0 + y < 0 \\ 0 & d_0 + y \geq 0 \end{cases} \tag{1}$$

where $d_0$ is the gap size, y is the relative displacement between the deck and roadbed, and $K_P$ is the pounding stiffness. According to past studies [34–36], a large $K_P$ value is usually assumed. In this analysis, the transverse stiffness of the deck is referred to define the $K_P$ value, which is set as $5.38 \times 106$ kN/m.

### 2.4. Artificial Seismic Excitation

Earthquake ground motions are stochastic loads, which are difficult to calculate accurately. The actual earthquake data is lacking in the structure address, so artificial ground motions are selected as the input ground motions to simulate random earthquake actions [37,38]. In this paper, artificial ground motions are generated from the famous SIMOKE code (Vanmarcke and Gasparini [13]) based on the design response spectrum, which is according to the specification for the seismic design of highway bridges of the Chinese design code [6]. The PGA of the design response spectrums ranges from 0.1 g to 0.6 g, and 10 artificial earthquake waves are generated for each design response spectrum to conduct the nonlinear time history analyses. The time step size of these artificial earthquake waves is 0.02 s, and the period is 16 s. Figure 4 shows an example of the comparison of the artificial response spectrum and the given design response spectrum for a damping ratio of 5%. The result shows that the seismic excitation well matches the given design response spectrum.

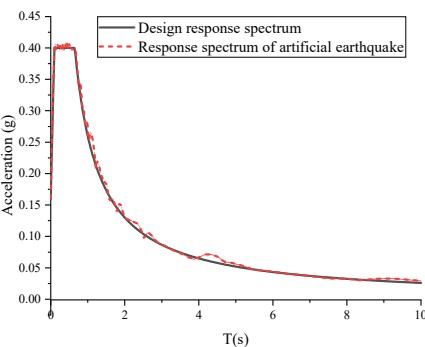

**Figure 4.** The response spectrum of the artificial ground motion.

## 3. Results and Observation

In order to study the dynamic behaviors of the prefabricated frame bridge under the ground motions with various intensities, seismic responses of the elastoplastic column-deck joints (EC-DJ) and the prestressed columns (PC) are investigated. Considering that the earthquake is a random process, so the maximum response of the structure is evaluated by random simulations, and the artificial seismic excitations are selected as ground motions input. In this section, the intensities of peak ground acceleration (PGA) are set as 0.1 g, 0.2 g, 0.3 g, 0.4 g, 0.5 g, and 0.6 g, respectively, to generate a total of six groups of design acceleration response spectra, and 10 artificial waves are generated for each design spectrum. Then the bending moment-curvature curves are extracted to analyze the energy consumption of the pounding effects. At last, the seismic intensity and the gap distance are investigated to explore the pounding effects on the seismic behaviors of the prefabricated frame bridge.

### 3.1. Seismic Response Analysis

Time history analysis can reflect the pounding effect on the prefabricated frame bridge clearly, so the responses of the EC-DJ and prestressed column (PC) are analyzed with considering pounding and without considering pounding. As is shown in Figures 5 and 6, the responses of EC-DJ and PC increased greatly at 6.6–7.8 s in the case of 0.3 g PGA, while the responses of the EC-DJ and PC increased a lot at 6.5–9.5 s in the case of 0.6 g PGA. As is shown in Figure 7, the pounding force surged at 5.8–7.0 s in the case of 0.3 g PGA. In the case of 0.6 g PGA, the pounding force surged from 5.8 s to 7.0 s. Moreover, pounding times in the case of 0.6 g PGA are more than that of 0.3 g PGA.

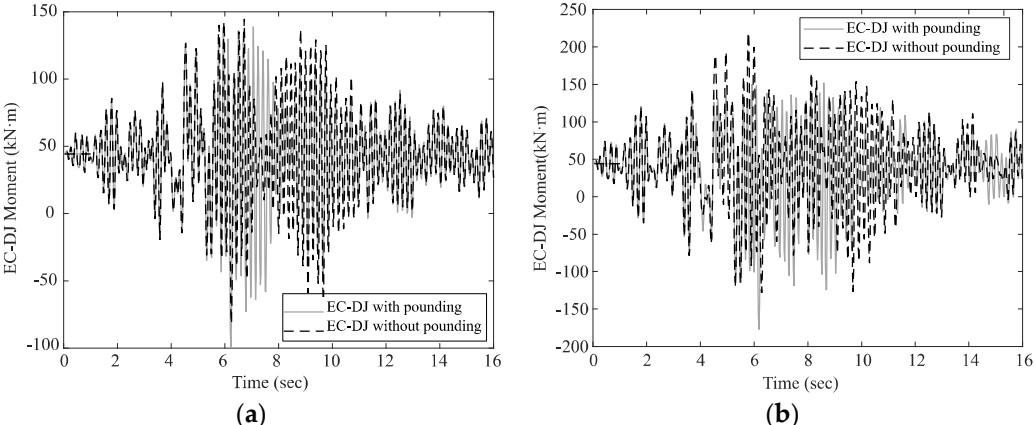

**Figure 5.** EC-DJ force. (**a**) EC-DJs force for 0.3 g PGA. (**b**) EC-DJs force for 0.6 g PGA.

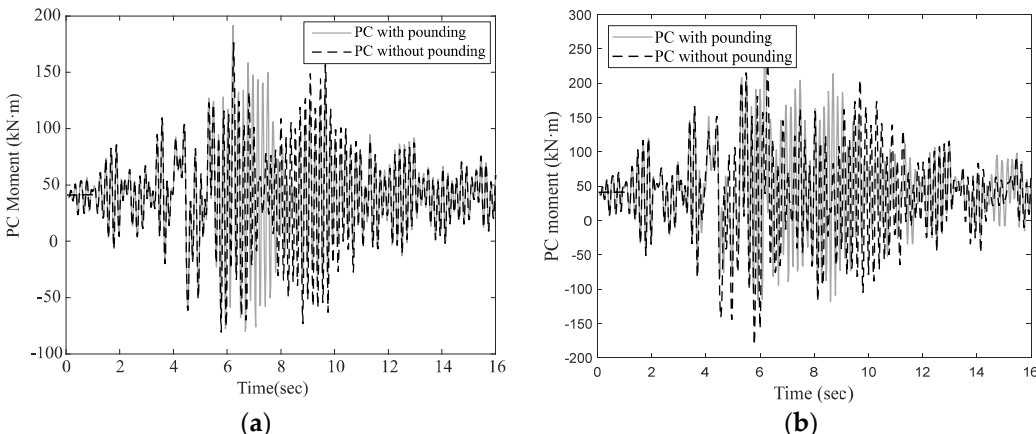

**Figure 6.** Prestressed columns force. (**a**) Prestressed columns force for 0.3 g PGA. (**b**) Prestressed columns force for 0.6 g PGA.

Figures 8 and 9 show the comparison of the force-displacement relationship of EC-DJ and PC for 0.3 g PGA and 0.6 g PGA. The larger the enclosed area, the more seismic energy is dissipated. The areas enclosed by the dotted curve (without pounding) are smaller than the solid curve (with pounding), obviously, especially when the PGA is 0.3 g.

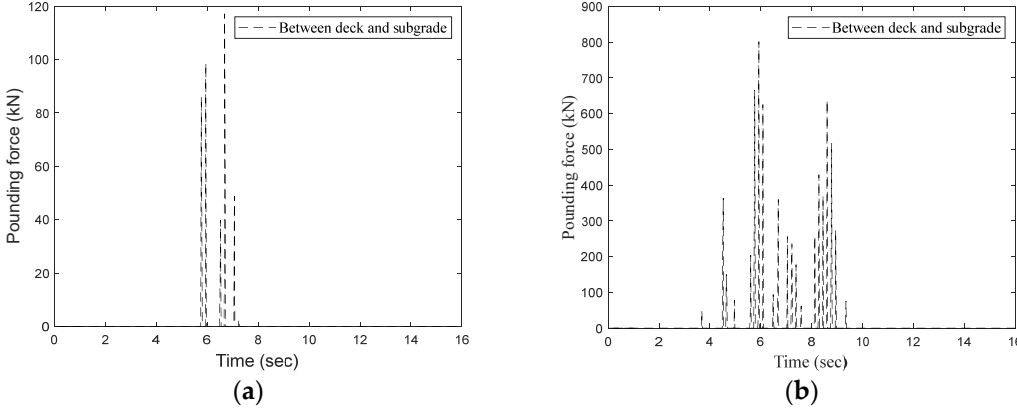

**Figure 7.** Pounding force. (**a**) Pounding force for 0.3 g PGA. (**b**) Pounding force for 0.6 g PGA.

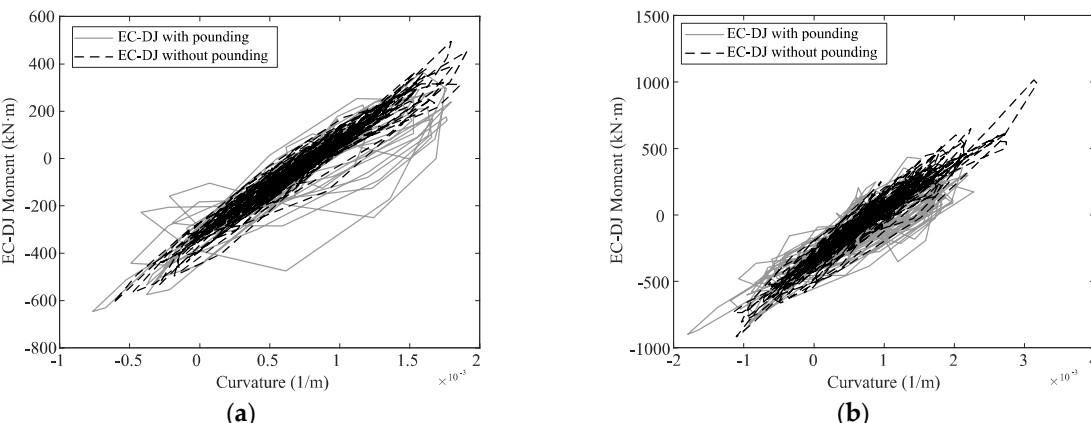

**Figure 8.** EC-DJ force-displacement relationship under different PGA. (**a**) EC-DJ force-displacement relationship for 0.3 g PGA. (**b**) EC-DJ force-displacement relationship for 0.6 g PGA.

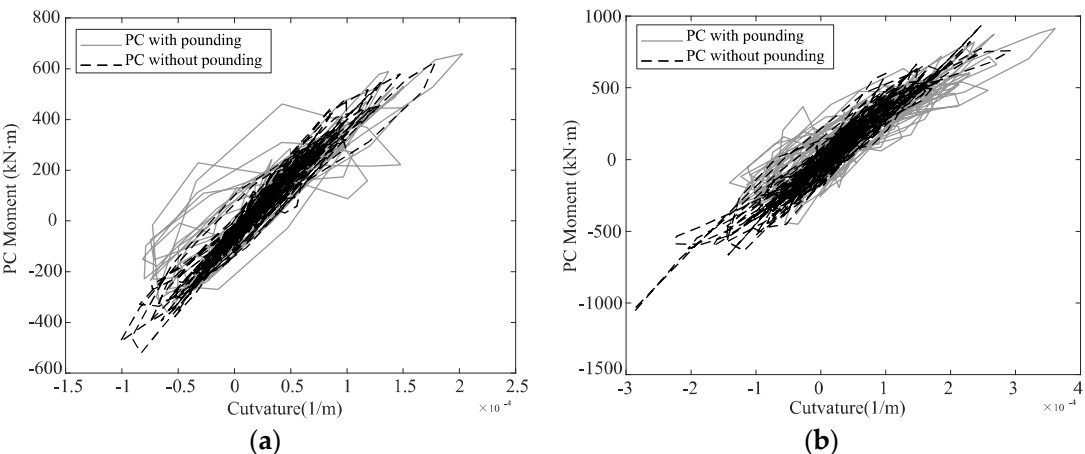

**Figure 9.** PC force-displacement relationship under different PGA. (**a**) PC force-displacement relationship for 0.3 g PGA. (**b**) PC force-displacement relationship for 0.6 g PGA.

Figures 10–13 show the bending moment-curvature relationship of the EC-DJ and the PC in different cases. The dotted line and the solid line nearly coincide when PGA is under 0.2 g, which indicates that pounding does not occur in those cases.

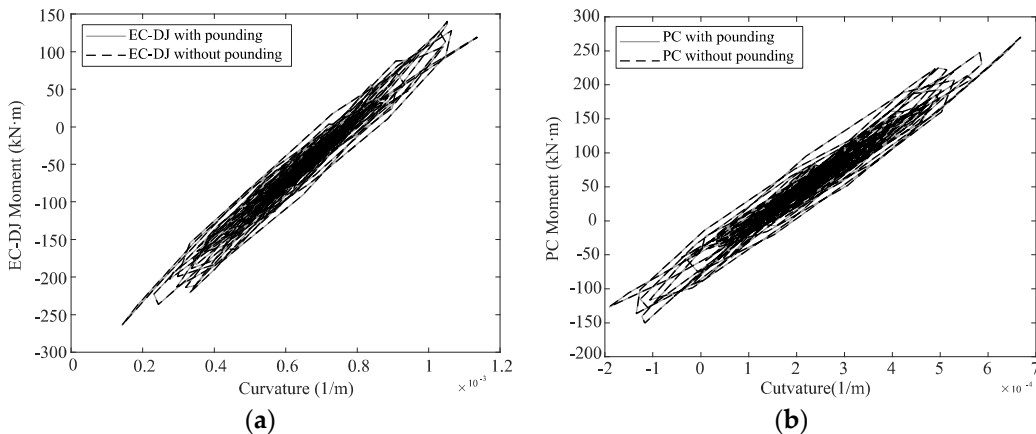

**Figure 10.** Seismic responses under 0.1 g PGA. (**a**) EC-DJ force-displacement relationship for 0.1 g PGA. (**b**) PC force-displacement relationship for 0.1 g PGA.

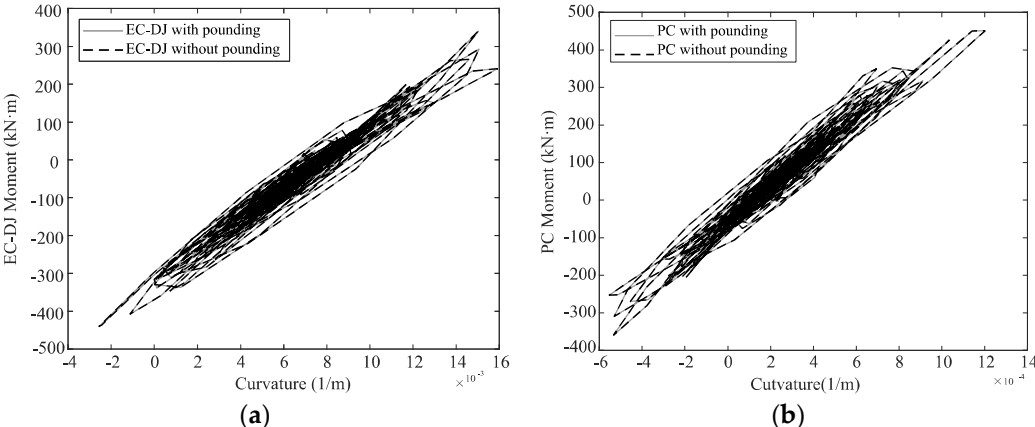

**Figure 11.** Seismic responses under 0.2 g PGA. (**a**) EC-DJ response for 0.2 g PGA. (**b**) PC response for 0.2 g PGA.

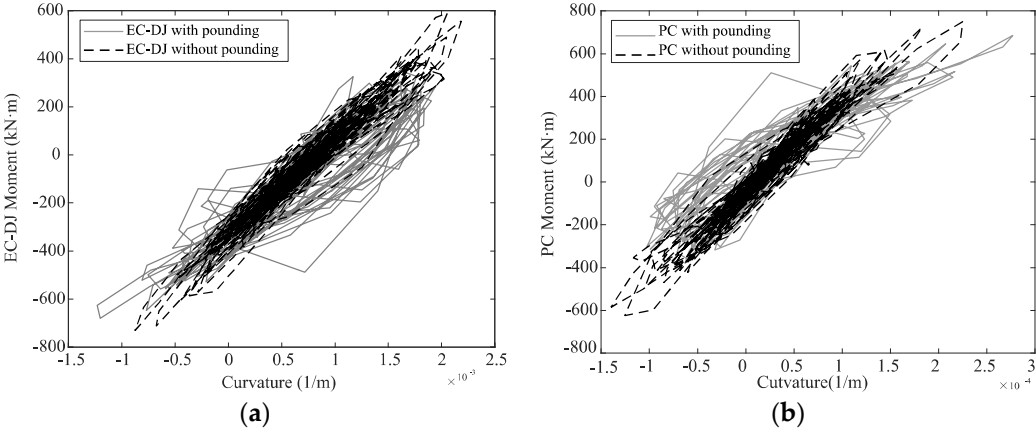

**Figure 12.** Seismic responses under 0.4 g PGA. (**a**) EC-DJ response for 0.4 g PGA. (**b**) PC response for 0.4 g PGA.

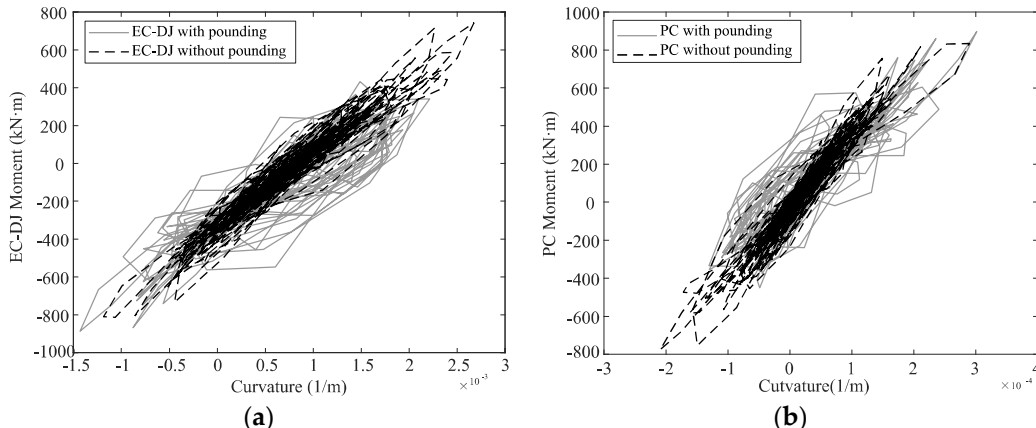

**Figure 13.** Seismic responses under 0.5 g PGA. (**a**) EC−DJ response for 0.5 g PGA. (**b**) PC response for 0.5 g PGA.

In the case of PGA is 0.3 g, 0.4 g, 0.5 g, and 0.6 g, compared with the enclosed areas of the dotted line and solid line, it can be found that the larger the PGA is, the less obvious the capacity of consuming seismic energy due to the pounding effect. Although much seismic energy can be consumed in the pounding process, it is limited to some extent.

When the PGA is not higher than 0.2 g, the EC-DJs and PCs are under the linear state, and no pounding occurs. When the PGA is higher than 0.2 g, the pounding times are more, and the pounding force is larger, which indicates that the pounding effects on the prefabricated frame bridge are more obvious. In this case, the EC-DJs and the PCs enter the nonlinear state but are recoverable, and some seismic energy is consumed in the pounding process.

### 3.2. Response Characteristics According to Various Gap Distances

Gap distance is a key factor that influences the pounding potential between the deck and the roadbed. In this section, seismic responses of the prefabricated frame bridge with different gap distances are recorded to analyze the pounding effect between the deck and the roadbed. To conduct the nonlinear time history analyses, six groups of artificial waves with different ground motion intensities are selected as ground motion input. The analysis, including the cases considering the pounding effect with the pounding gaps, is set as 0 mm, 10 mm, 20 mm, 30 mm, 40 mm, 50 mm, and 60 mm, and the pounding effect is ignored.

To describe the analysis results clearly, the direction of the deck close to the original roadbed is defined as the positive direction, and the reverse direction is defined as the negative direction. The maximum positive transverse deformation and the minimum negative transverse deformation for each case are shown in Figures 14 and 15.

The maximum positive deformation reflects the pounding between the deck and the original roadbed occurs or not. When the maximum positive deformation is larger than the gap distance, it can be concluded that the pounding exists. According to the comparative analysis of Figure 14, when the gap distance ranges from 0 to 20 mm, pounding exists under each seismic intensity; When the gap distance ranges from 30 to 40 mm, pounding exists in case the seismic intensity is greater than 0.3 g PGA; When the gap distance ranges from 50~60 mm, pounding exists in case the seismic intensity is not lower than 0.5 g PGA.

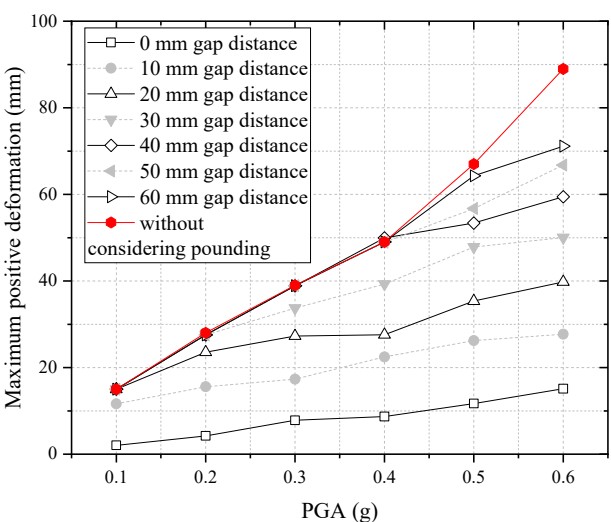

**Figure 14.** Maximum positive deformation of the deck.

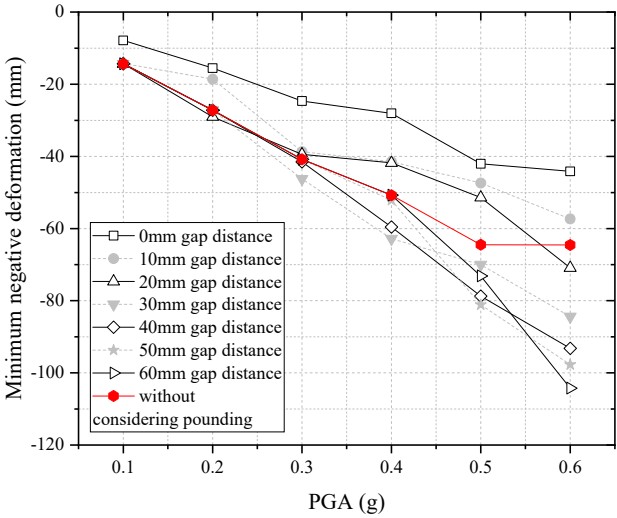

**Figure 15.** Minimum negative deformation of the deck.

The minimum negative deformation reflects the movement trend of the deck away from the original roadbed soil. When the negative deformation is large enough, the bridge deck may be separated from the original roadbed, resulting in the cantilever stress state of the bridge deck. The minimum negative deformation increases with the increasing seismic intensity, as can be seen in Figure 15. When the gap distance is not smaller than 40 mm, the minimum negative deformation of the deck is approximate for each seismic intensity; when the gap distance is smaller than 40 mm, it is clear that the minimum negative deformation increases with the increasing gap distance. When the pounding effect is considered, the minimum negative deformation value will increase by up to 200% at most.

In this study, it is found that the value of the minimum negative deformation reaches 104 mm. However, in the process of design and construction, the length of the deck overlapping on the original roadbed is designed as 1 m, which is large enough to prevent the stress state of the deck from changing, and the safety of the superstructure is guaranteed to some extent.

The mean responses of the EC-DJs and PCs are calculated from the bending moment during 10 nonlinear time history analyses for different PGAs, which are plotted in Figure 16. Because the EC-DJs and the prestressed columns are connected nearly rigidly, the responses of the two components are approximate.

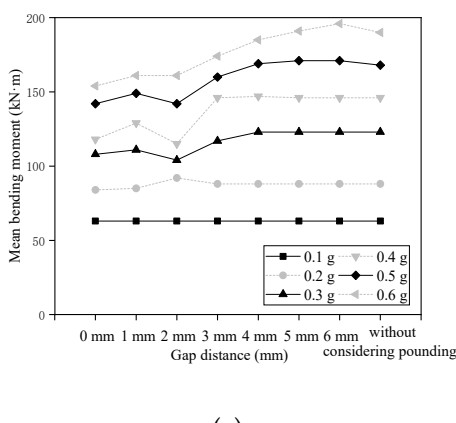
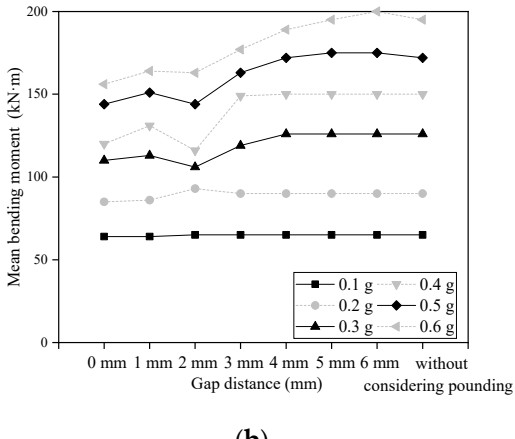

**Figure 16.** Mean response of the components. (**a**) Mean response of the EC−DJ. (**b**) Mean response of the PC.

The deck is connected with piers firmly by the elastoplastic column-deck joint (EC-DJ), so the seismic responses of the components will be decreased. The seismic responses decrease obviously when the pounding is severe. Original roadbed soil plays a role in restricting the deck deformation, but this function depends on the soil stiffness. When the PGA is low and the gap distance is large, the pounding potential is low. When the PGA is too high and the gap distance is small, the soil will be destroyed due to pounding, and the original roadbed cannot limit the deck deformation or consume more seismic energy. For these two cases, the pounding effect can be ignored.

According to the comparative analysis, it can be seen from Figure 16 that when the seismic intensity is low, such as 0.1 g PGA and 0.2 g PGA, whether considering the pounding effect or not has little impact on the seismic response of EC-DJs and prestressed columns. Moreover, when the PGA is not lower than 0.2 g, the seismic response of the EC-DJs and prestressed columns is decreased with the increasing seismic intensity.

Compared to the system without considering the pounding effect, the seismic response of the system considering the pounding effect is reduced by 5% at most, with PGA is 0.1 g and 0.2 g; the seismic response is reduced by 15.6% at most with PGA is 0.3 g; the seismic response is reduced by 21.6% at most with PGA is 0.4 g; the seismic response is reduced by 15.5% at most with PGA is 0.5 g; the seismic response is reduced by 19.2% at most with PGA is 0.6 g.

## 4. Conclusions

Prefabricated frame bridges are emerging structures designed to solve the problem of difficult land acquisition in highway expansion and reconstruction. As shown in Figure 2, this new structure has been applied in the Wu-He expressway since 2019, which is proven to be practical, environmentally friendly, and economical.

Considering that the deck of the prefabricated frame bridge is adjacent to the original roadbed, the pounding between the deck and roadbed probably occurs under the earthquake ground motions, which will influence the seismic response to some extent. In this paper, the pounding effect is simulated by the Kelvin pounding model, and the pounding effect on the seismic response is investigated, accounting for two key parameters, including the gap distance and the seismic intensity. Then the pounding effect is assessed by comparing the seismic response of the two models, with and without consideration of pounding. The following trends can be concluded from the results:

(1) Pounding effects do not always exist under seismic excitation. When the pounding occurs, some seismic energy will be consumed, and the seismic response of the components will decrease;

(2) The smaller the gap distance and the higher intensity of the seismic excitation, the higher the pounding potential is. When the PGA is not higher than 0.2 g, the pounding effect can be ignored;

(3) The minimum negative deformation value will reach 104 mm at most when the pounding effect is considered, which is still much lower than the overlapping length between the bridge deck and the original roadbed, so the safety of the superstructure is ensured due to the cantilever state of the bridge deck is not occur.

The seismic response law and damage mechanism of prefabricated frame bridges are relatively complex, so theoretical analysis is not enough. However, experiments are lacking in this paper. For future studies, the corresponding shaking table test and pseudo-static test should be added to further verify and enrich the theoretical research results.

**Author Contributions:** Conceptualization, L.C.; Methodology, L.C.; Software, Y.W.; Validation, Y.W. and R.Z.; Formal analysis, T.S.; Investigation, J.Z. and T.S.; Resources, Y.Z.; Data curation, Y.W., J.Z., Y.Z. and T.S.; Writing—original draft, Y.W.; Writing—review & editing, L.C. and T.S.; Visualization, J.Z. and R.Z.; Supervision, L.C.; Project administration, J.Z. All authors have read and agreed to the published version of the manuscript.

**Funding:** This research was funded by the Anhui Province Natural Science Foundation of China grant number 2208085ME151.

**Institutional Review Board Statement:** Not applicable.

**Informed Consent Statement:** Informed consent was obtained from all subjects involved in the study.

**Data Availability Statement:** Data is unavailable due to privacy.

**Conflicts of Interest:** The authors declare no conflict of interest.

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
