# Peer review of "Effects of the Transverse Deck-Roadbed Pounding on the Seismic Behaviors of the Prefabricated Frame Bridge"

_sustainability, doi:10.3390/su15021554_

Round 1

Reviewer 1 Report

The paper aims to investigate the seismic performance of prefabricated bridge frames under seismic excitations. The pounding effect's influence is explored through increasing intensity underground motions. It is apparent that the smaller the gap distance will yield higher pounding force under higher seismic intensity. The authors also emphasize the reduction in the seismic response of components when the gap is smaller than 40mm as the gap dissipates the energy and decreases the demand on the columns.

The paper has many editorial discrepancies as the manuscript lack coherence and is difficult to understand.

The authors should go through my comments added in red in the attached draft and resubmit for review.   

Reviewer 2 Report

The pounding effects on the prefabricated frame-bridge are investigated based on the pounding forces and the components seismic response. The test results showed that the pounding effects not always exist under the seismic excitation. The smaller the gap distance and the higher intensity of the seismic excitation, the higher the pounding potential is. The safety of the superstructure is guaranteed to some extent considering the value of the minimum negative deformation. This research could provide evidence for the practical application of prefabricated frame-bridge in highway expansion and reconstruction. It should be accepted for the Journal after the following comments are carefully considered.

1.       It would be better if the introduction had ended with a description of the practical engineering relevance of the study.

2.       The content of “Fig.3(c), (d), (e), (f)” in page 4 and lines 119 does not match the presentation in Figure 3. Please check and revise.

3.       The “Chouw N, et al [18] maintained” is not in the same format as “Guo A et al [13], present”. These similar contents should also be checked and corrected.

4.       The font size in the coordinate axes and legends in the Figure 6 (a) do not match in Figure 6 (b). Please check and revise.

5.       Please carefully check the format of the references, e.g. whether the names of the authors are consistent, whether they contain [J], etc.

Reviewer 3 Report

The conducted work “Effects of the Transverse Deck-roadbed Pounding on the Seismic Behaviors of the Prefabricated Frame-bridge” is good. However, following comments should be addressed to further improve the paper:

A. GENERAL COMMENTS FOR IMPROVING PAPER ON OVERALL BASIS

1.      Explicitly mention the novelty and research significance of current work in last paragraph of introduction section with emphasis on scientific soundness. Also, add recent relevant literature review more from 2022 papers in introduction section as there is no paper cited from 2022.

2.      Avoid paragraph of few (2-4) sentences throughout the manuscript, particularly in results and discussions sections e.g. lines 180-183, 184-187, 188-191, etc.

3.      Results should be further elaborated with scientific reasoning.

4.      A separate brief section (explaining the relevance of this research for practical implementation) may be added before conclusion section.

5.      Conclusions should be reflection of obtained results with scientific soundness. Conclusions are little long; these should be to the point as obtained from results. Closing remarks should be added at the end of conclusion section keeping in mind all conclusive bullet points.

6.      English Language should be improved throughout the manuscript, e.g. lines 279-280

B. SPECIFIC COMMENTS FOR IMPROVING FOCUSSED RESEARCH

1.      Figures 9 to 14: all these figures have same caption i.e. Force-displacement relationship. It is better to have one figure with sub-captions.

2.      Line 270 “the seismic response of the components will decrease”: by how much it will be decreased and why?

3.      Line 275 “pounding effect can be ignored”: Why it can be ignored? It needs justification with scientific reasoning.

Round 2

Reviewer 1 Report

The Authors have addressed all the comments and I don’t have additional comment.